# Topological Analysis for Sleep Apnea Detection: Project Proposal

Dylan Li, Yeap Yun Ting

April 27th, 2026

## 1 Introduction

Sleep apnea is one of the most widespread sleep disorders worldwide, affecting more than one billion individuals [1]. However, its often subtle symptoms, combined with the complexity and cost of clinical diagnosis, make large-scale detection challenging. Polysomnography, long viewed as the standard diagnostic procedure, is increasingly questioned for its high resource requirements and relatively poor scalability.

**Gaps identified:** These limitations have motivated the development of alternative automated approaches based on machine learning and deep learning, which typically rely on a reduced set of physiological signals for patient-level sleep apnea detection. Although these methods often achieve strong performance on standard metrics, they still fail to fully model the complex relationships inherent in sleep-related data. In particular, they do not adequately capture recurrent geometric structures of the signals, thereby overlooking information embedded in extended-range temporal patterns.

**Novelty/contribution:** In this work, we propose a first framework based on topological data analysis for sleep apnea detection. Specifically, we leverage persistent homology, a tool that extracts robust features from structurally similar inputs, enabling a more comprehensive characterization of both local and global patterns in ECG signals. Moreover, this work aims to investigate whether cardiac data alone can provide sufficient information for reliable sleep apnea detection

## 2 Motivation

Obstructive Sleep Apnea (OSA), commonly referred to as sleep apnea, is a sleep-related breathing disorder characterized by partial or complete obstruction of the upper airway, leading to recurrent interruptions of normal sleep patterns. Although its clinical manifestations are often less overt than those of other diseases, OSA can have significant consequences, including excessive daytime sleepiness, cognitive and behavioural impairments, an increased risk of domestic and traffic accidents, as well as elevated cardiovascular mortality if not treated[2].

Polysomnography (PSG) is currently considered the gold standard for the diagnosis of sleep apnea. It typically requires patient monitoring over one or two nights using a range of sensors, including electroencephalography (EEG), electrocardiography (ECG), pulse oximetry and electrooculography (EOG), among others. However, this approach is frequently criticized for its high cost in terms of both time and resources. In addition, the unfamiliar clinical environment and equipment may affect patient sleep quality, potentially leading to altered or less representative recordings[3].

Automated sleep scoring algorithms have recently emerged as promising alternatives to PSG. These approaches typically focus on a limited set of easily accessible, noise-robust, and clinically informative signals, most notably ECG and peripheral oxygen saturation ($SpO_2$) obtained via pulse oximetry. Machine learning and deep learning techniques have been widely applied in this context [10]-[17], with most models relying on physiological features locally extracted from time series data. However, such approaches often fail to capture the recurrent patterns inherent to sleep dynamics[4].

To address these limitations, we propose a novel framework based on the topological analysis of key physiological signals for sleep apnea detection. By leveraging global geometric structure within each temporal window, this approach aims to enhance the accuracy and robustness of sleep apnea detection compared to the existing baselines.

## 3 Background

Detection of sleep apnea can take different forms depending on the target objective. Some studies aim to determine whether an individual suffers from OSA, framing the problem as a binary classification task, while others focus on identifying the specific temporal segments associated with apnea events, which corresponds to a sequence labeling task. Given that topological data analysis has demonstrated strong potential for capturing local dynamical structures

and recurrent patterns in the signal[5], we restrict the scope of this work to patient-level binary classification.

Historically, OSA has been clinically diagnosed using polysomnography. However, its routine use has increasingly been questioned due to its resource-intensive nature. PSG requires significant time, financial, and clinical resources, and may be unnecessarily complex when the sole objective is the detection of sleep apnea. Indeed, many of the recorded signals are primarily intended for the diagnosis of a broader range of sleep disorders and are not strictly required for OSA detection [6]. Consequently, a growing body of work has focused on machine learning and deep learning approaches relying on a reduced set of physiological signals—most commonly ECG and $SpO_2$—that are more directly related to apnea events. These approaches have demonstrated promising performance while enabling more scalable and less intrusive diagnostic solutions [10]-[17]. However, these algorithms struggle to leverage the information contained in the local geometrical structures of the signal. This motivates the use of alternative representations which can handle better these patterns, such as topological data analysis.

TDA is a framework for studying the shape of data by extracting geometric and topological structures from complex signals. In this project, we focus on persistent homology, a core TDA tool that identifies topological features such as connected components and cycles across multiple scales. These features are summarized using persistence diagrams, which encode the lifespan of each topological structure as the scale varies. From these diagrams, we can also compute distances such as bottleneck or Wasserstein distances, which provide quantitative measures of similarity between signals and are used for classification tasks on physiological data such as ECG or $SpO_2$[7][8][9].

# 4 Related work

- **ECG-based approaches in the time domain**

  This category includes methods that directly exploit the temporal structure of single-lead ECG signals. Most approaches rely on either baseline machine learning classification models [10] or on neural networks [11]. These methods are effective at capturing short-term physiological changes associated with apneic events, such as variations in R–R intervals or local morphological patterns. However, they mainly rely on cardiac dynamics alone, which do not fully capture the multi-system nature of sleep apnea.

- **ECG-based approaches in the frequency domain**

  These methods analyze ECG signals in the spectral domain to capture frequency-related characteristics associated with sleep apnea. Techniques such as power spectral density estimation [12] or bispectral [13] are used to extract features that reflect autonomic nervous system activity and periodic physiological changes. By transforming the signal into the frequency domain, these approaches can highlight patterns that are less visible in the time domain. However, they often lose fine-grained temporal information and still rely on relatively local or stationary assumptions about the signal.

- **Automated approaches using $SpO_2$ signals**

  These methods use oxygen saturation signals to detect sleep apnea based on desaturation events and related physiological changes. Both classical machine learning [14] and deep learning [15] models have been successfully applied in this setting, benefiting from the direct physiological link between oxygen desaturation and apneic events. However, these methods remain lar gely event-based and sensitive to noise, and they mainly reflect delayed oxygen desaturation responses rather than the full underlying respiratory dynamics of sleep apnea.

- **Hybrid ECG + $SpO_2$ approaches**

  Hybrid approaches combine ECG and $SpO_2$ signals to improve robustness and classification performance, using their complementarity for a more complete and accurate characterization of sleep apnea. Although these methods often achieve better results than single-modality approaches [16] [17], their exploration remains limited by the scarcity of high-quality datasets collecting simultaneously both signals.

- **Topological Data Analysis for sleep and physiological signals**

  Recent works have investigated Topological Data Analysis as an alternative framework for time series analysis [5]. In particular, studies based on persistent homology have shown its ability to capture the recurrent geometric structures of physiological signals, with successful applications to sleep-related tasks such as sleep-wake detection [19] and sleep stage analysis [20]. Moreover, sleep apnea has also been studied using TDA in previous work [21], although these contributions mainly focus on clustering phenotypes in already suspected or diagnosed patient populations. In this context, applying topological methods directly to sleep apnea detection appears to be less commonly explored.

# 5 Challenges

- **Data challenges**

The dataset comes with several practical challenges that need to be addressed before any modeling. First, ECG signals are inherently noisy, due to motion artifacts, sensor imperfections, and external interference, which makes proper filtering essential. In addition, sleep signals are highly non-stationary, meaning their statistical properties change over time depending on sleep stages and transient events like apnea. This makes the choice of time window particularly important, as it directly impacts the quality of the extracted features. Finally, the dataset is typically imbalanced, with fewer apnea events compared to normal segments, which can introduce bias in the learning process if not handled carefully.

- **Approach challenges**

  From a methodological point of view, using Topological Data Analysis (TDA) involves several important choices. In particular, we need to define how the data is represented (i.e if we work within the feature space, with embeddings. . . ) and which topological structures are used . These choices directly affect the features that are extracted. In addition, the method depends on several parameters, such as embedding dimensions or filtration scales, which need to be tuned carefully. Another key challenge is interpretability: while TDA provides rich geometric features, linking them to meaningful physiological insights is not always straightforward.

- **Physiological challenges**

  In this project, we choose to focus only on single-lead ECG signals, with the idea that this may reduce noise and simplify the overall approach. However, this comes with important limitations. Sleep apnea is not directly observable from cardiac data and must instead be inferred from indirect physiological responses, such as changes in heart rate variability. This means that the signal may only partially reflect apnea events. Moreover, similar patterns can be caused by other factors such as stress or movements, making the task more ambiguous. Overall, this makes it difficult to clearly distinguish apnea-related patterns from other sources of variation.

# 6   Objectives

The objective of this project is to develop a lightweight, interpretable obstructive sleep apnea (OSA) detection pipeline based on topological domain time series analysis of single-lead ECG signals, targeting competitive performance on the public Apnea-ECG database to support future deployment on consumer wearable devices.

To achieve this core goal, the project will first quantify the discriminative performance of topological features extracted from RR interval sequences for OSA detection, before constructing the full Topo-ECG hybrid pipeline that fuses these topological features and traditional heart rate variability (HRV) features via a lightweight attention-based bidirectional long short-term memory (BiLSTM) or encoder-only Time Series Transformer (TST) module, while systematically comparing performance across different feature combinations and classification head configurations. Finally, the project will validate the interpretability of the complete pipeline by visualizing the correlation between extracted persistent homology features and OSA events.

# 7   Methodology

## 7.1   Overall pipeline

The proposed Topo-ECG OSA detection pipeline follows a modular, interpretable architecture designed to address the three core challenges identified earlier: signal noise, non-stationary sleep dynamics, and the difficulty of extracting apnea patterns from single-lead ECG data. Raw single-lead ECG input first passes through a preprocessing module that cleans noise, extracts RR interval sequences, and computes traditional heart rate variability (HRV) features. Cleaned RR intervals are simultaneously passed to a topological feature extraction module, which uses phase space reconstruction and persistent homology to generate global geometric feature vectors that capture long-term temporal patterns missed by traditional feature sets.

The two parallel feature streams (traditional HRV and topological features) are fed into an attention-based fusion layer that dynamically weights features based on their discriminative value for OSA detection, before being passed to a lightweight classification head (either BiLSTM or Time Series Transformer) that generates per-minute apnea predictions. Per-minute predictions are aggregated to the patient level using the standard clinical AHI cutoff of 10% apneic minutes to produce a final binary OSA diagnosis. This modular design enables straightforward ablation testing to isolate the contribution of individual components and supports simple deployment on edge wearable devices due to its low computational overhead.

## 7.2   Data collection and preprocessing

The preprocessing workflow contains five sequential steps:

- Structured loading of raw single-lead ECG recording from the Apnea-ECG database.

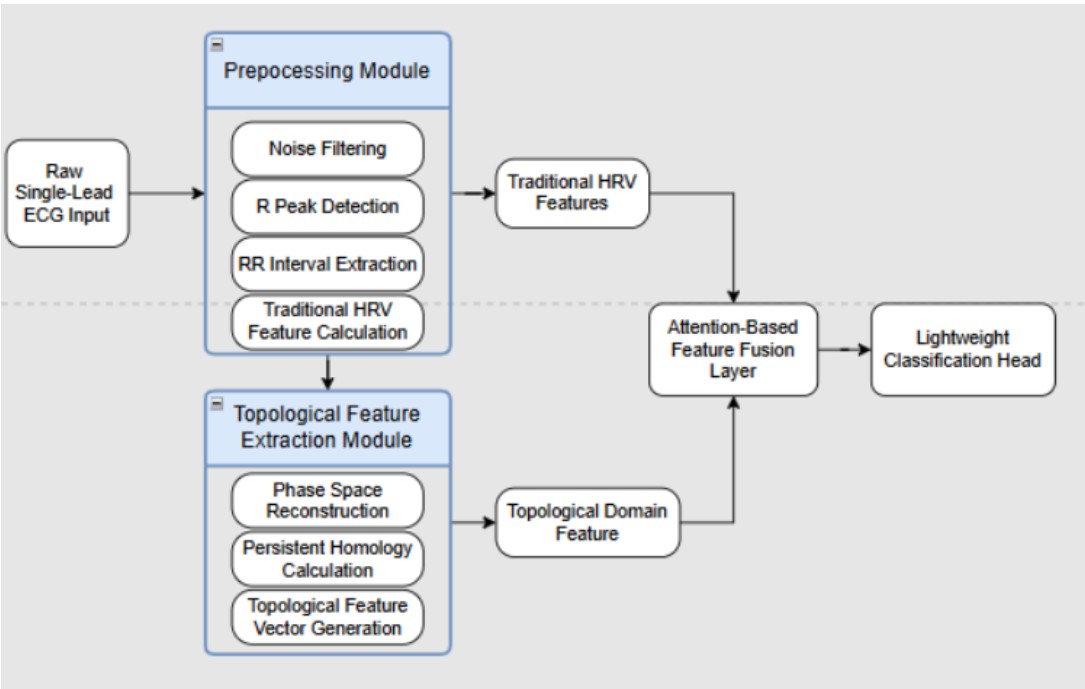

Figure 1: Overall pipeline of our study

- Multi-stage noise filtering to eliminate artifacts from motion interference, power line noise and baseline wander, ensuring signal quality for subsequent analysis.

- R peak detection to extract RR interval sequences representing the time intervals between consecutive heartbeats. An additional outlier filtering step is applied to extracted RR intervals, removing physiologically impossible values outside the 300ms–2000ms range; gaps of <3 consecutive missing intervals are imputed via linear interpolation, while windows with >3 missing RR values are discarded entirely to avoid feature corruption.

- Fixed-length 1-minute window alignment, consistent with the dataset annotation schema.

- Extraction of traditional HRV features across time and frequency domains as the baseline feature set for performance comparison.

## 7.3 Core model design

### 7.3.1 Topological feature extraction

The topological feature extraction module implements three core steps:

- Phase space reconstruction of cleaned RR interval sequences using Taken's embedding theorem, transforming the 1-dimensional time series into a high-dimensional state space that captures underlying dynamic patterns.

- Persistent homology calculation to quantify topological invariants (connected components, loops) of the reconstructed phase space across multiple scales.

- Generation of fixed-length topological feature vectors from the resulting persistence diagrams, suitable for input into downstream classification models.

### 7.3.2 Feature fusion and classification module

The module implements deep learning for classification:

- All input features (traditional HRV and TDA-derived) are first standardized to 0 mean and unit variance, fit exclusively on training fold data to eliminate cross-validation leakage, before being passed to the attention-based fusion layer. The attention-based feature fusion layer then dynamically assigns weights to topological and traditional HRV features based on their discriminative importance for OSA detection.

- Two optional lightweight classification heads: a bidirectional long short-term memory (BiLSTM) network optimized for sequential pattern learning, and a lightweight Time Series Transformer (TST) with only an encoder component, supporting flexible trade-offs between computational cost and detection performance.

## 7.4 Training strategy

The model training framework employs weighted cross-entropy loss to address the class imbalance in the dataset and use the AdamW optimizer with standard hyperparameter settings to optimize model parameters. All models are trained for 50 epochs with early stopping to prevent overfitting, and performance is validated using 5-fold cross-validation following subject-independent partitioning to ensure generalizable results.

## 7.5 Evaluation plan

The model performance is quantified using four key metrics: overall accuracy, F1-score, area under the receiver operating characteristic curve (AUC-ROC), and sensitivity, with a particular focus on high sensitivity to minimize missed diagnosis of OSA events. Per-minute window predictions are aggregated to the patient level using a clinically aligned threshold: a patient is classified as OSA-positive if 10% of their recorded windows are labelled as apnea, matching the standard clinical apnea-hypopnea index (AHI) cutoff for mild OSA. All core metrics are reported at both the per-window and patient level.

A comprehensive ablation study will be conducted to quantify the contribution of each pipeline component, including the following experimental conditions:

- Random Forest classifier with only traditional HRV features, establishing the classical machine learning baseline for conventional features.

- Random Forest classifier with only TDA features, verifying the independent discriminative capacity of topological features with classical ML models.

- Random Forest classifier with fused TDA + traditional HRV features, validating the feature fusion effect with classical ML models.

- BiLSTM head with only traditional HRV features as input, establishing the baseline performance of conventional OSA detection methods.

- BiLSTM head with only topological domain features as input, verifying the independent discriminative capacity of TDA-derived features.

- BiLSTM head with fused features as input (full proposed pipeline), verifying the feature fusion effect with BiLSTM model.

- TST head with only traditional HRV features as input, evaluating the performance of Transformer architectures without topological features.

- TST head with only topological domain features as input, verifying the independent discriminative capacity of TDA-derived features for Transformer architectures

- TST head with fused features as input (full proposed pipeline), verifying the feature fusion effect with TST model.

Additional robustness testing will be conducted by evaluating model performance on ECG signals down-sampled to 25Hz and 50Hz, plus 0.1mV and 0.2mV Gaussian white noise to mimic motion artifacts recorded by consumer wearables, verifying the pipeline's suitability for low-quality signals captured by consumer wearable devices.

## 7.6 Interpretability analysis

Two complementary interpretability analysis approaches will be implemented to ensure the pipeline outputs are clinically interpretable:

- Visualization of persistence diagrams for normal and OSA-containing 1-minute windows, explicitly demonstrating the differences in topological patterns between the two categories.

- Calculation of the attention weight distribution of the classification module, quantifying the contribution of different topological and traditional features to OSA detection decisions, providing clinically readable evidence supporting the model's outputs.

# 8 Dataset

This project uses the publicly available Apnea-ECG Database [22] from PhysioNet [23] , which is the most widely used benchmark dataset for ECG-based OSA detection research.

- **Sample size** The dataset contains 70 single-lead ECG recordings from overnight polysomnography (PSG) studies of subjects suspected of having sleep apnea. Each recording varies in length from slightly less than 7 hours to nearly 10 hours, with a sampling frequency of 100 Hz. The dataset includes a total of 34,313 1-minute ECG windows. Each minute was classified as either a "non-apnea minute" or an "apnea minute".

- **Official train/test split** The dataset is officially divided into a training set of 35 recordings and a test set of 35 recordings. Each recording includes a continuous digitized ECG signal, a set of apnea annotations (derived by human experts on the basis of simultaneously recorded respiration and related signals), and a set of machine-generated QRS annotations (in which all beats regardless of type have been labeled normal).

- **Annotation rules** All ground truth annotations are provided by experienced sleep clinicians based on simultaneous PSG recordings, following the standard AASM (American Academy of Sleep Medicine) guidelines for apnea and hypopnea scoring. This project uses only the official "apn" reference annotation files as the ground truth source for all model training and evaluation, with no supplementary or alternative label sources used.

# 9 Project schedule

| Week | Dates | Tasks | Details | Lead / Support |
|---|---|---|---|---|
| 1 | Apr 30–May 6 | Dataset validation; preprocessing; baseline model | Verify PhysioNet Apnea-ECG data. Preprocess (R peaks, filtering, 1-min windows, HRV). Train RF baseline; report Acc, F1, AUC, Sens. | Dylan / Yeap |
| 2 | May 7–13 | Phase reconstruction; persistent homology; validation | Takens embedding on RR intervals. Compute persistence diagrams (components, loops). Build features and test correlation with OSA. | Yeap / Dylan |
| 3 | May 14–20 | Fusion layer; classifier; training | Attention-based fusion (HRV + topo). Implement BiLSTM/TST. Train with weighted CE, 5-fold CV, early stopping. | Dylan / Yeap |
| 4 | May 21–27 | Evaluation; interpretability; report | Ablation + robustness (noise, downsampling). Analyze persistence diagrams + attention weights. Final report + code. | Yeap / Dylan |

Table 1: Project Timeline

# 10 Progress so far

- **Completed**

  So far, our work has focused on reviewing related projects and exploring different approaches for our study. In particular, we looked at the previously used machine learning and deep learning algorithms to evaluate which ones could be relevant for topological data analysis. We also worked on understanding the dataset, including the nature of the signals and their annotations, while identifying possible weaknesses that could affect our results. Finally, since TDA was a relatively new topic for both of us, we spent some time learning its main theoretical concepts as well as its practical applications.

- **Ongoing**

  We are currently working on data preprocessing. After loading the signals and verifying the completeness of both the data and their annotations, we moved on to building the processing pipeline. This includes R-peak detection, noise filtering, the choice of sliding windows, and the extraction of HRV features. Once this step is completed, we plan to train a baseline model (e.g., a Random Forest) for the OSA detection task. This will allow us to check that the preprocessing has been carried out correctly before moving on to the topological analysis itself.

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
