# OpenReview forum: "Topological Analysis for Sleep Apnea Detection: Project Proposal"
_tsinghua.edu.cn/THU/2026/Spring/ANM — THU 2026 Spring ANM Submission_

### Official Review · Reviewer_oz5g · 2026-05-13

**Rating:** 7
**Confidence:** 3

**Summary:**

Current diagnostic standards for sleep apnea are expensive and impractical, and existing ML/DL approaches still miss recurring patterns in physiological signals over time. To address both problems, this project proposes Topo-ECG, a framework that uses topological data analysis (TDA), specifically persistent homology, to detect obstructive sleep apnea (OSA) using only a single ECG signal.

The pipeline starts by cleaning the raw ECG signal and extracting RR intervals, the time gaps between heartbeats, which are then divided into one-minute segments. Each segment is processed in two ways simultaneously. One stream computes traditional heart rate variability (HRV) features, while the other applies Takens embedding and persistent homology to capture the geometric shape of the signal. Both streams are then combined through an attention layer that automatically learns which features are most useful before being passed into a lightweight classifier, either a BiLSTM or a Time Series Transformer, to predict whether each minute contains an apnea event.

The model is evaluated on the publicly available Apnea-ECG database from PhysioNet. A structured ablation study compares different model and feature combinations to isolate each component's contribution, and robustness is further tested on degraded signals to confirm the pipeline works reliably under real-world wearable conditions.

**Strengths:**

- Apply a topological analysis on OSA detection: captures global geometric and recurring temporal patterns that traditional HRV features and standard ML/DL methods typically miss
- Using ECG only: Far simpler and less intrusive compared to polysomnography, which requires multiple sensors simultaneously
- Attention-based feature fusion: Combines topological and traditional HRV features, allowing the model to leverage the best of both representations rather than relying on either alone

**Weaknesses:**

- Single signal limitation: Relying solely on single-lead ECG means apnea-related patterns may be confused with other physiological events such as stress or movement, making it difficult to reliably distinguish true apnea events from noise
- TDA computational cost: Persistent homology is generally expensive to compute, particularly for long overnight recordings, yet the paper does not address how this scales in practice or whether it meets real-time wearable constraints
- Parameter sensitivity: TDA requires careful tuning of key parameters such as embedding dimensions and filtration scales, which adds complexity to the pipeline and may reduce reproducibility when applied to different datasets or patient populations. Therefore, the author may need to add a careful study in this area.

**Questions:**

- Was any optimization or approximation used to speed up the computation?
- Was the pipeline ever tested on actual wearable hardware, or only on standard computers?

---

### Official Review · Reviewer_yuDL · 2026-05-13

**Rating:** 8
**Confidence:** 3

**Summary:**

This proposal explores using topological data analysis to improve sleep apnea detection from ECG signals. The idea is to capture patterns in the data that standard methods might miss by combining topological features with traditional heart rate features. The authors propose a full pipeline and plan to evaluate it on a standard dataset, including tests to understand how each component contributes to the overall performance.

**Strengths:**

1) The proposal is built around a well-motivated and conceptually strong research idea, leveraging topological data analysis as a novel perspective for modeling complex physiological signals.
2) Authors provide decent overview of the existing work in Section 4.
3) The experimental design is carefully structured and includes a thorough ablation study, which is valuable for isolating the contribution of individual components and strengthening the validity of the conclusions.

**Weaknesses:**

1) Work [4] does not exist or not properly cited.
2) According to authors, existing works considering TDA "mainly focus on clustering phenotypes in already suspected or diagnosed patient populations". While it is also noted that main limitation of applying TDA to considered data is difficulty "to clearly distinguish apnea-related patterns from other sources of variation". This may explain why application of TDA to detection is not as promising as TDA of already suspected patients, and there is no information provided on how the authors are planning to overcome this specific difficulty.
3) In 7.3.1, authors refer to Taken’s embedding theorem as a method for phase-space reconstruction. In fact, the theorem itself provides conditions to make reconstruction possible, but (to the best knowledge of the reviewer) doesn't do the reconstruction itself.

---

### Official Review · Reviewer_LNs2 · 2026-05-14

**Rating:** 7
**Confidence:** 3

**Summary:**

Te proposal aims to detect sleep apnea from single-lead ECG by extracting topological features via persistent homology applied to the phase space of RR intervals, fusing them with traditional HRV features through an attention mechanism, and classifying the result using a BiLSTM or Transformer head.  The idea seems innovative and falls within time series analysis, providing a well thought out plan, but still having some parts of the temporal modelling unclear

**Strengths:**

•	The project addresses a real-world clinical problem using a public benchmark dataset and well-defined evaluation metrics aligned with clinical standards (AHI threshold).
•	The modular pipeline design enables systematic ablation testing, making it possible to isolate the contribution of topological features.
•	The proposal explicitly identifies a gap, in existing methods often missing recurrent geometric structures in signals, and proposes a well defined way in to address it.

**Weaknesses:**

•	It seems unclear whether the classification head processes a sequence of minute-level feature vectors or classifies each minute independently. If each minute is classified independently, the claim to capture “extended-range temporal patterns” may not be entirely supported.
•	Persistent homology on a single-window point cloud discards temporal ordering within the window, weakening the argument that it captures long-term dynamics better than standard nonlinear HRV features (e.g., entropy, DFA).
•	The attention-based fusion mechanism is described only vaguely, making it difficult to judge its provided value.

**Questions:**

1.	Does the classification head see a sequence of minute-level feature vectors, and if so, what would be the time horizon?
2.	What is the exact architecture of the attention-based fusion layer?

---

### Official Review · Reviewer_6ZRA · 2026-05-15

**Rating:** 8
**Confidence:** 3

**Summary:**

The research aims to detect Obstructive Sleep Apnea (OSA) by applying Topological Data Analysis (TDA), specifically persistent homology, to single-lead ECG signals. The methodology extracts both topological and traditional Heart Rate Variability (HRV) features from RR intervals, fusing them via an attention mechanism into a Bidirectional LSTM or Time Series Transformer. The end goal is an interpretable, lightweight model tailored for consumer wearables.

**Strengths:**

Integrating Takens' embedding theorem to reconstruct the phase space of 1D RR intervals directly addresses the failure of local time-domain models to capture extended-range temporal patterns and non-stationary sleep dynamics. The ablation study is well-structured, systematically isolating the independent discriminative capacity of traditional HRV versus topological features across a classical machine learning baseline (Random Forest) and deep learning models (BiLSTM, TST). The robustness evaluation specifically targets the signal degradation expected in wearable edge devices by introducing 0.1mV and 0.2mV Gaussian white noise and down-sampling the 100Hz signals to 25Hz and 50Hz.

**Weaknesses:**

From a physiological standpoint, relying solely on single-lead ECG limits the model to indirect autonomic responses (HRV). Because apneic events share cardiac signatures with general arousals, movements, or stress, the classification boundary remains highly ambiguous without respiratory or $SpO_{2}$ data. While the proposal acknowledges the sensitivity of TDA parameters, it completely omits the optimization strategy. Specifically, computing persistence diagrams requires selecting an embedding dimension and time delay for the phase space reconstruction, as well as defining the distance metric and maximum filtration scale for the Vietoris-Rips (or similar) complex, none of which are detailed. The strategy for managing the dataset's class imbalance relies entirely on a weighted cross-entropy loss. Given the use of data-hungry architectures like a Time Series Transformer , weighting the loss might be insufficient to prevent the model from overfitting to the majority "normal" class within the 34,313 available windows.

**Questions:**

What specific heuristic or grid-search strategy will be used to determine the embedding dimension and time delay parameters required for Takens' phase space reconstruction?

The pipeline handles missing RR intervals via linear interpolation for gaps of fewer than three beats. How do you ensure this synthetic interpolation does not create artifactual cycles or connected components during the persistent homology calculation?

Given the highly non-stationary nature of sleep signals , do you anticipate that standardizing the topological feature vectors to zero mean and unit variance  might obscure the absolute geometric scale of the phase space, which could be relevant for distinguishing apnea?

---

### Official Review · Reviewer_cTfQ · 2026-05-17

**Rating:** 8
**Confidence:** 3

**Summary:**

This proposal studies sleep apnea detection from single-lead ECG signals using Topological Data Analysis, especially persistent homology. The authors argue that standard ECG-based machine learning methods mainly rely on local time/frequency features and may miss recurrent geometric structures in heart-rate dynamics. Their proposed pipeline extracts RR intervals from ECG, computes traditional HRV features, applies phase-space reconstruction and persistent homology to obtain topological features, then fuses HRV and TDA features using an attention-based module before classification with either a BiLSTM or a lightweight Time Series Transformer. The final predictions are made at the per-minute level and then aggregated to the patient level. The project uses the PhysioNet Apnea-ECG database and proposes ablations comparing HRV-only, TDA-only, and fused-feature variants.

**Strengths:**

1. The proposal addresses an important and practical problem and does this clearly with clinically motivated reasons

2. The pipeline is reasonably well specified. he diagram on page 4 also makes the proposed architecture easy to understand: ECG input is split into a traditional HRV branch and a topological feature branch, then fused before classification.

3. The proposed ablation plan is strong and well structured. The authors propose comparisons between HRV-only, TDA-only, and fused HRV+TDA features, using both classical Random Forest models and neural classification heads, thus directly testingwhether TDA adds value beyond standard HRV features.

4. The proposal shows good awareness of leakage and evaluation issues.

5. The authors analyzed the existing work in detail and presented it clearly

**Weaknesses:**

1. The proposed use of persistent homology on 1-minute RR windows is plausible but may be statistically fragile. Each window contains a limited number of beats, and delay embedding further reduces the number of available points. This may make higher-dimensional topological features, especially loops, sensitive to noise and parameter choices.

2. TDA features from delay-embedded RR intervals may capture variability, periodicity, and irregularity. But traditional HRV features already capture many of these properties, so there's a huge risk that TDA features won't add information beyond HRV.

3. Because the TDA module is built on RR intervals, its reliability depends strongly on R-peak detection quality. Missed or false R peaks could create artificial geometric structures in the embedded point cloud. The robustness analysis should therefore consider beat-detection errors, not only Gaussian noise or downsampling.

**Questions:**

1. What exact topological representation will be used to convert persistence diagrams into fixed-length features?

2. Will the primary task be per-minute apnea detection, patient-level OSA diagnosis, or both?

3. The proposal states that interpretability will be evaluated through persistence-diagram visualization and attention-weight analysis. However, the authors also acknowledge that linking TDA features to physiological insights is not straightforward. How will they verify that the observed topological differences correspond to meaningful apnea-related physiology rather than noise, preprocessing artifacts, or generic HRV variability?

---

### Official Review · Reviewer_6VGe · 2026-05-17

**Rating:** 7
**Confidence:** 4

**Summary:**

This proposal presents a framework for obstructive sleep apnea detection using Topological Data Analysis (TDA), particularly persistent homology, applied to single-lead ECG signals. The authors propose combining topological features extracted from RR interval sequences with conventional HRV features and lightweight sequential models such as BiLSTM and Time Series Transformers. The work is clinically relevant, well-motivated, and aims to provide an interpretable and wearable-friendly ECG-only solution for sleep apnea detection.

**Strengths:**

Interesting and relatively novel application of TDA for ECG-based sleep apnea detection.
Strong motivation and practical relevance for wearable and low-cost diagnosis.
Well-structured and comprehensive methodology.
Includes robustness testing, ablation studies, and interpretability analysis.
Good awareness of challenges such as signal noise, non-stationarity, and class imbalance.
Clear evaluation plan using standard benchmarks and metrics.

**Weaknesses:**

Novelty claim is somewhat overstated since prior TDA-based sleep analysis work already exists.
Important technical details about the TDA pipeline are missing (feature vectorization, embedding parameter selection, filtration strategy).
Potential overfitting risk due to the small dataset size relative to the proposed model complexity.
Computational cost and deployment feasibility of persistent homology are not sufficiently analyzed.
Some references are weak or non-peer-reviewed sources.
Clinical justification for the patient-level thresholding strategy needs clearer explanation.

**Questions:**

How are persistence diagrams converted into fixed-length feature vectors?
How are Takens embedding parameters selected?
What is the computational overhead of persistent homology extraction?
Why are BiLSTM/TST models necessary after handcrafted topological features are extracted?
How is patient-level data leakage avoided during preprocessing and cross-validation?
How will the authors ensure fair comparison between HRV-only, TDA-only, and fused models?

---

### Official Review · Reviewer_o7GZ · 2026-05-18

**Rating:** 8
**Confidence:** 3

**Summary:**

The authors propose Topo-ECG, a pipeline for obstructive sleep apnea (OSA) detection from single-lead ECG that combines traditional heart rate variability (HRV) features with topological features extracted via persistent homology. The pipeline reconstructs the phase space of RR sequences with Takens embedding, computes persistence diagrams over the resulting point cloud, and fuses the topological features with HRV through an attention layer. Classification is done by either a BiLSTM or a Time Series Transformer head. Evaluation is on the public Apnea-ECG database from PhysioNet, with a comprehensive ablation across feature combinations and classifier heads, plus robustness tests on downsampled and noise-corrupted signals.

**Strengths:**

- The pipeline is modular and with strong ablation plan. Nine experimental conditions covering Random Forest, BiLSTM, and TST with each feature combination is more than most course proposals attempt, and it will produce clean evidence on whether topological features actually add value over HRV alone.
- Methodological hygiene is good. Feature standardization is fit only on training folds, cross-validation is subject-independent, and patient-level aggregation uses the clinically-aligned 10% AHI cutoff for mild OSA.
- The robustness plan (downsampling to 25 and 50 Hz, Gaussian noise injection) is well-motivated by the wearable deployment practical goal and goes beyond typical course project scope.
- The interpretability plan is concrete on the topological side. Visualizing persistence diagrams for normal vs OSA windows is a sensible diagnostic and could surface clinically meaningful patterns

**Weaknesses:**

- The TDA pipeline is underspecified on the parameter side: embedding dimension, time delay, filtration choice, and persistence-diagram vectorization are all left open, and these choices materially affect the results.
- There is a real risk that TDA features overlap with what HRV already measures (variability, periodicity, irregularity). The ablation is designed to test this empirically, but the proposal does not articulate which topological signal is supposed to be uniquely captured beyond HRV.
- Computational cost of persistent homology is not analyzed, which could conflict with the stated wearable deployment goal.

**Questions:**

- Which vectorization of persistence diagrams will you use, and how will you choose takens embedding parameters?
- What is the per-window computational cost of the persistent homology step, and is it compatible with on-device inference?
- Could the ablation include a TDA + nonlinear HRV (entropy, DFA) condition, to test whether TDA adds information beyond nonlinear HRV specifically?

---

### Official Review · Reviewer_huPm · 2026-05-18

**Rating:** 6
**Confidence:** 4

**Summary:**

[AI Review] This paper proposes a framework using Topological Data Analysis (TDA) for sleep apnea detection on the Apnea-ECG dataset. The review evaluates a class project proposal by Dylan Li and Yeap Yun Ting, identifying critical issues with novelty claims and evaluation protocols while noting strong writing and experimental design.

**Strengths:**

1. Well-written with professional structure and presentation.
2. Honest and transparent discussion of anticipated challenges.
3. Comprehensive 9-condition ablation design that allows systematic evaluation of contributions.
4. Thoughtful preprocessing pipeline design.
5. Strong clinical grounding, including appropriate use of AHI threshold and sensitivity focus for clinical relevance.

**Weaknesses:**

1. False novelty claim: The paper claims to be the 'first framework based on TDA for sleep apnea detection,' but Yan et al. (2022) already developed a similar approach on the same Apnea-ECG dataset and is not cited.
2. Evaluation protocol conflict: Unclear whether the paper uses 5-fold cross-validation or the official 35/35 train/test split on Apnea-ECG, which could inflate reported results.
3. Questionable additive value of TDA features over established nonlinear HRV measures (sample entropy, Poincaré plots, DFA), which may capture redundant information.
4. Underspecified persistence diagram vectorization: No method is specified for converting persistence diagrams to feature vectors for downstream models.
5. Unrealistic 4-week timeline, particularly Week 2 scope requiring authors (who are TDA beginners) to learn TDA and implement a full pipeline simultaneously.

**Questions:**

1. How do you address the unacknowledged prior work by Yan et al. (2022) that applies TDA to sleep apnea detection on the same dataset?
2. Which evaluation protocol will be used (5-fold CV or official train/test split), and how will reproducibility and fair comparison be ensured?
3. What specific persistence diagram vectorization method will be used (persistence images, landscapes, or summary statistics)?
4. How will you determine that persistent homology features provide information beyond what existing nonlinear HRV measures capture?
5. What are the quantitative success criteria (target accuracy, F1 score, or other metrics) for this project?

---

### Official Review · Reviewer_w5HS · 2026-05-19

**Rating:** 7
**Confidence:** 3

**Summary:**

This proposal introduces a sleep apnea detection pipeline using single-lead ECG signals. The method extracts RR intervals, computes traditional HRV features, and combines them with topological features from persistent homology. These features are fused and classified using either a BiLSTM or Time Series Transformer, with evaluation planned on the PhysioNet Apnea-ECG dataset.

**Strengths:**

The project tackles an important clinical problem with a practical ECG-only approach. The pipeline is clear, modular, and well suited for ablation studies. The proposed comparisons between HRV-only, TDA-only, and fused features are strong, and the robustness tests with noise and downsampling are relevant for wearable deployment.

**Weaknesses:**

The TDA part is still underspecified: embedding dimension, delay, filtration choice, and persistence diagram vectorization are not clearly defined. The novelty claim should also be softened, since TDA has already been used in sleep-related physiological analysis. It is also unclear whether TDA features will add information beyond standard HRV features, and the evaluation protocol should clarify whether the official train/test split, 5-fold cross-validation, or both will be used.

**Questions:**

-How will persistence diagrams be converted into fixed-length features?
-How will embedding dimension and time delay be selected?
-Will the main evaluation use the official Apnea-ECG split, 5-fold CV, or both?
-What does TDA capture beyond traditional HRV features?
-What is the computational cost of persistent homology per window?